# N-type and P-type series integrated hydrogel thermoelectric cells for low-grade heat harvesting

Jiafu Shen, Xi Huang, Yu Dai, Xiaojin Zhang ⊕ ✉ & Fan Xia ⊕ ✉

Low-grade heat is abundant and ubiquitous, but it is generally discarded due to the lack of cost-effective recovery technologies. Ion thermoelectric cells are an affordable and straightforward approach of converting low-grade heat into usable electricity for sustainable power. Despite their potential, ion thermoelectric cells face challenges such as limited Seebeck coefficient and required series integration. Here, we demonstrate that the N-type and P-type conversion of ion thermoelectric cells can be achieved through the phase transition of temperature-sensitive hydrogel containing the triiodide/iodide redox couple. Through the strong interaction between the hydrophobic region of the hydrogel and triiodide, the hydrophobic side selectively captures triiodide and the hydrophilic side repels triiodide, raising the concentration difference of triiodide and thereby increasing the Seebeck coefficient. Specifically, the Seebeck coefficient of the N-type ion thermoelectric cells is $7.7\,\mathrm{mV\,K^{-1}}$, and the Seebeck coefficient of P-type ion thermoelectric cells is $-6.3\,\mathrm{mV\,K^{-1}}$ ($\Delta T = 15\,\mathrm{K}$). By connecting 10 pairs of the N-type and P-type ion thermoelectric cells, we achieve a voltage of 1.8 V and an output power of 85 μW, surpassing the reported triiodide/iodide-based ion thermoelectric cells. Our work proposes a phase transition strategy for the N-P conversion of ion thermoelectric cells, and highlights the prospect of series integrated hydrogel ion thermoelectric cells for low-grade heat harvesting.

Low-grade heat, such as solar heat[1], low-temperature industrial waste heat[2] and human body heat[3], is widely present in nature but is generally discarded. Efficiently harvesting and harnessing low-grade heat is crucial for sustainable development[4]. In recent years, thermoelectric conversion technologies, such as organic Rankine cycle[5], Kalina cycle[6], thermomagnetic effect[7] and thermoelectric effect[8], have been developed to directly convert heat into electricity. Among these, the thermoelectric effect has been widely studied[9]. Conventional electronic thermoelectric (e-TE) materials have low Seebeck coefficient ($S$) ($\mathrm{\mu V\,K^{-1}}$) and high cost[10]. In contrast, ion thermoelectric materials including ion thermoelectric supercapacitors (i-TESCs)[11–13] and ion thermoelectric cells (i-TECs)[14], have received great attention due to

their $\mathrm{mV\,K^{-1}}$ $S$ and low cost. The i-TECs generate electricity through a temperature difference driven redox reaction, which is a sustainable energy source[15]. Some i-TECs with different redox couples have been developed[16]. Despite the progress made, the i-TECs still face challenges such as limited $S$ and required series integration. For example, the absolute $S$ of $I_3^-/I^-$, $Fe(CN)_6^{-3}/Fe(CN)_6^{-4}$ and $Fe^{3+}/Fe^{2+}$ i-TECs is $0.5$–$0.8\,\mathrm{mV\,K^{-1}}$ (ref. 17), $1.4\,\mathrm{mV\,K^{-1}}$ (ref. 18) and $1.04\,\mathrm{mV\,K^{-1}}$ (ref. 19), respectively.

The $S$ is determined by the formula $S = (\Delta S + \Delta C)/nF$, where $\Delta S$ and $\Delta C$ are related to the solvation structure entropy and concentration entropy of redox couple, respectively[20]. Increasing $\Delta S$ can significantly boost the $S$[21]. To increase the $\Delta S$ of $[Fe(CN)_6]^{3-}/[Fe(CN)_6]^{4-}$, strong

State Key Laboratory of Biogeology and Environmental Geology, Engineering Research Center of Nano-Geomaterials of Ministry of Education, Faculty of Materials Science and Chemistry, China University of Geosciences, Wuhan, China. ✉e-mail: zhangxj@cug.edu.cn; xiafan@cug.edu.cn

chaotropic cations (guanidinium) and highly soluble amide derivatives (urea) were introduced[22]. Due to ionic bonding interaction, $\Delta S$ of $[Fe(CN)_6]^{3-}/[Fe(CN)_6]^{4-}$ increased, raising the $S$ from 1.4 mV K$^{-1}$ to 4.2 mV K$^{-1}$. Another strategy involves regulating $\Delta C$ to boost the $S$. Guanidinium cations can enhance $\Delta C$ between the hot and cold sides by selectively inducing the crystallization of $Fe(CN)_6^{4-}$. A high $S$ of 3.7 mV K$^{-1}$ and a Carnot-relative efficiency ($\eta_r$) of 11.1% are obtained[15]. Although the efficiency of single cell continues to improve, the limited voltage generated by single cell hinders high voltage demands.

P-type and N-type series integration can effectively increase the thermal voltage of i-TECs[23]. A high voltage of 2.05 V was achieved by connecting 32 pairs of $Fe^{3+}/Fe^{2+}$ i-TEC and $[Fe(CN)_6]^{3-}/[Fe(CN)_6]^{4-}$ i-TEC in series[19]. The P-type and N-type i-TECs are typically different redox couples, which may cause cross-infection[24]. For example, $Fe(CN)_6^{-3}/Fe(CN)_6^{-4}$ exhibits good stability under neutral and alkaline conditions, but produces highly toxic hydrogen cyanide under acidic conditions,

making it incompatible with acidic i-TECs such as $Fe^{3+}/Fe^{2+}$ (ref. 25). To address this issue, hydrophilic and hydrophobic convertible nanogel was introduced into redox couple triiodide/iodide ($I_3^-/I^-$)[17]. At high temperature, the nanogel captures free $I_3^-$ in the solution, and then diffuses to low temperature through Brownian motion, becoming hydrophilic and releasing ions. The P–N conversion occurs in liquid i-TECs and requires ion carriers to achieve concentration difference of redox ions under temperature difference[26]. The liquid i-TECs are prone to leakage and have poor mechanical performance[27]. The hydrogel i-TECs can avoid these problems[28]. Temperature-sensitive hydrogels have different hydrophilicity/hydrophobicity under temperature difference[29]. The hydrophobic region attracts $I_3^-$, and the hydrophilic region repels $I_3^-$, thereby enhancing the concentration difference of redox ions.

Here, we report the N−P conversion of hydrogel i-TECs with same redox couple, and demonstrate series integration of hydrogel i-TECs for low-grade heat harvesting (Fig. 1a). Different from the previously

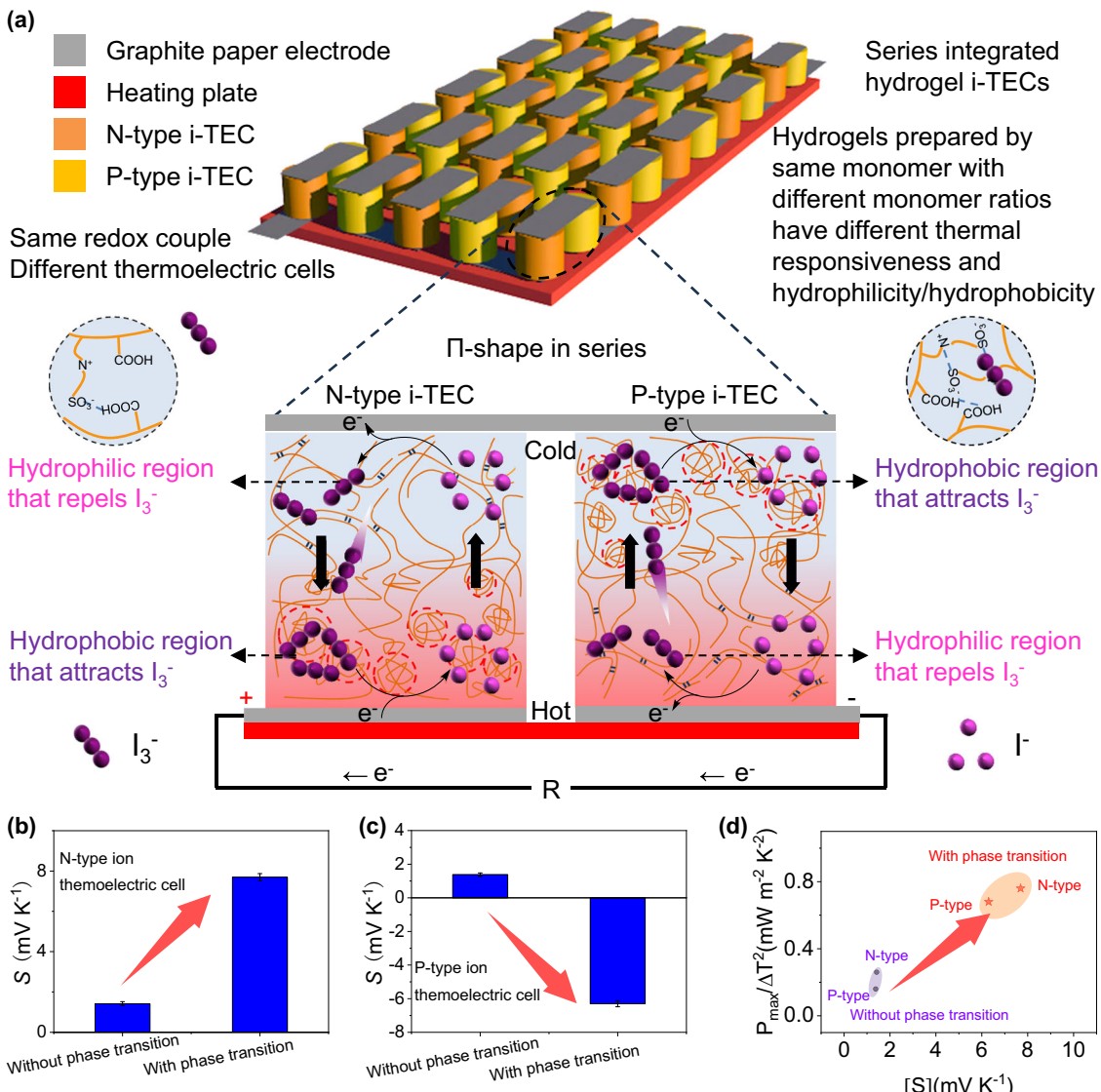

**Fig. 1 | Series integrated hydrogel ion thermoelectric cells (i-TECs) and phase transition enhanced thermoelectric performance. a** The N-type and P-type i-TECs are sandwiched between two flexible substrates and coupled alternatively in a Π-shape using graphite paper electrode. The expanded graphic depicts the mechanism of the N-type and P-type i-TECs. When the hot-side temperature ($T_h$) exceeds the phase transition temperature ($T_p$), the hydrophobic hot side of the N-type i-TEC attracts triiodide ($I_3^-$), and the hydrophilic cold side repels $I_3^-$. The

hydrophilic hot side of the P-type i-TEC repels $I_3^-$, and the hydrophobic cold side attracts $I_3^-$. **b** Phase transition enhanced Seebeck coefficient of N-type i-TEC. **c** Phase transition induced thermoelectric type conversion. A detailed explanation is shown in Supplementary Note 1. **d** Phase transition enhanced normalized instantaneous power density ($P_{max}/\Delta T^2$). The error bars were calculated using the standard deviation of the measured Seebeck coefficient.

reported i-TECs, the concentration difference of redox couples in our i-TECs is caused by the hydrogel phase transition, which does not depend on the carrier. The hydrogel has both lower critical solution temperature (LCST) and upper critical solution temperature (UCST) by adjusting the monomer ratio[30]. This will generate different hydrophilicity/hydrophobicity at the same temperature, resulting in different attraction/repulsion effects on $I_3^-$. Therefore, the hydrogel can be used to prepare the N-type and P-type i-TECs. Through the strong interaction between the hydrophobic region of the hydrogel and $I_3^-$, the concentration difference of redox ions is enhanced, thereby increasing the $S$ (Fig. 1b), converting N-type i-TEC to P-type i-TEC (Fig. 1c), and improving normalized instantaneous power density (Fig. 1d). The $S$ of the N-type i-TEC is 7.7 mV K$^{-1}$, and the $S$ of P-type i-TEC is −6.3 mV K$^{-1}$ ($\Delta T = 15$ K). A flexible thermoelectric cell by connecting ten pairs of N−P i-TECs reaches a voltage of 1.8 V ($\Delta T = 35$ K), promoting the application of wearable electronics.

## Results

### Preparation and characterization of hydrogel i-TECs

The redox couple $I_3^-/I^-$ is widely used in i-TECs. Generally, the oxidation reaction ($3I^- - 2e^- \rightarrow I_3^-$) occurs at the cold side, and the generated electrons flow through the external circuit to the hot side for the reduction reaction ($I_3^- + 2e^- \rightarrow 3I^-$)[31]. Compared to $I^-$, $I_3^-$ is more hydrophobic due to its lower level of charge density and is more prone to hydrophobic interaction[32]. Based on the characteristics of $I_3^-$, we construct hydrogel i-TECs with different hydrophilicity/hydrophobicity.

The hydrogel was prepared by one-pot copolymerization of methacrylic acid (MAA) and 3-dimethyl(methacryloyloxyethyl) ammonium propanesulfonate (DMAPS) with 2-hydroxy-2-methylpropiophenone as the photoinitiator (Fig. S1). Carboxylic, sulfonic and ammonium cations are associated with hydrogen bonding and ionic interaction[33,34]. The methyl group of MAA forms hydrophobic interaction[35]. By adjusting the mass ratio of the two monomers, hydrogels with different temperature-sensitive properties can be obtained[30]. The hydrogel i-TEC was prepared by immersing the hydrogel in the $I_3^-/I^-$ solution (Fig. S1).

When the temperature exceeds its phase transition point, the N-type i-TEC becomes hydrophobic, and the P-type i-TEC becomes hydrophilic. Under a certain temperature difference (hot-side temperature ($T_h$) > phase transition temperature ($T_p$), cold-side temperature ($T_c$) < $T_p$), the N-type i-TEC is hydrophobic at the hot side and hydrophilic at the cold side, and the P-type i-TEC is exactly the reverse. This leads to the cold side repelling $I_3^-$ and the hot side attracting $I_3^-$ of the N-type i-TEC, and the P-type i-TEC exhibits the opposite effect, with the hot side repelling $I_3^-$ and the cold side attracting $I_3^-$.

Mechanical properties of hydrogel i-TECs were evaluated through tensile and compression tests. The hydrogels and i-TECs have strong tensile capacity, allowing for stretching exceeding 250% strain (Fig. S2). Adding redox couple does not result in significant changes in compression property of hydrogels (Fig. S3). The tensile cycles (Fig. S4) and compression cycles (Fig. S5) of hydrogel i-TECs were further examined, and the results show that 80% of the performance could still be retained after three cycles.

### Temperature-sensitive and thermoelectric properties of hydrogel i-TECs

When the mass ratio of MAA and DMAPS ($m_{MAA}:m_{DMAPS}$) changes from 2:1 to 5:2, the LCST increases from 49 °C to 64 °C (Fig. 2a). Further increasing the mass ratio, the LCST exceeds 75 °C or even higher. If the mass of DMAPS exceeds MAA, the hydrogel shows UCST characteristics. When $m_{MAA}:m_{DMAPS}$ changes from 2:3 to 1:3, the UCST decreases from 63 °C to 32 °C (Fig. 2b). When $m_{MAA}:m_{DMAPS}$ is 2:1, UCST is 48 °C. If the N-type and P-type i-TECs want to achieve series integration, the required temperature should be consistent. Based on this consideration, we chose $m_{MAA}:m_{DMAPS} = 2:1$ and $m_{MAA}:m_{DMAPS} = 1:2$ to prepare

the N-type and P-type i-TECs, as their phase transition temperatures are essentially similar. At 25 °C, the N-type i-TEC shows good transmittance. When heated at 60 °C for 1 min, it turns opaque (Fig. 2c). The P-type i-TEC changes from opaque to transparent when heated from 25 °C to 60 °C. The thermal response is swift, indicating good reversibility. We conducted cyclic test on the transmittance of i-TECs at 25 °C and 60 °C. The i-TECs exhibit rapid reversible changes in transmittance (Fig. S6), implying their exceptional stability and reproducibility.

The $S$ of hydrogel i-TECs was measured using homemade equipment with graphite paper as electrode (Fig. S7). In the N-type i-TEC, the reduction reaction ($I_3^- + 2e^- \rightarrow 3I^-$) occurs at the hot side, and the oxidation reaction ($3I^- - 2e^- \rightarrow I_3^-$) occurs at the cold side (Fig. 2d). The cold-side temperature was maintained at 25 °C, and the open-circuit voltage was measured while slowly heating the hot side. The N-type i-TEC with different concentrations of $I_3^-$ was tested. The open-circuit voltage increases with the increase of the hot-side temperature (Fig. 2e). After the phase transition, the $S$ of the N-type i-TEC increases from 1.4 mV K$^{-1}$ to 3.5 mV K$^{-1}$ (Fig. 2f).

The oxidation and reduction reactions at both sides of the P-type i-TEC are opposite to those of the N-type i-TEC (Fig. 2g). The open-circuit voltage first increases and then decreases with the increase of the hot-side temperature, reaching its peaks at 45 °C (Fig. 2h). This result indicates that the hydrogel phase transition leads to the conversion from N-type to P-type. The phase transition temperature of hydrogel i-TECs (48 °C) closes the N−P conversion temperature (45 °C). When the $I_3^-$ concentration is 5 mM, the N-type and P-type i-TECs reach their maximum $S$ (Fig. 2i). Unless otherwise specified, the N-type i-TEC ($m_{MAA}:m_{DMAPS} = 2:1$) containing 5 mM $I_3^-$ and the P-type i-TEC ($m_{MAA}:m_{DMAPS} = 1:2$) containing 5 mM $I_3^-$ were used in the following experiments.

### Mechanism of enhanced thermoelectric properties

The $I_3^-$ and $I^-$ concentrations in the hydrogel i-TECs were monitored in real-time using in-situ Raman spectroscopy (Fig. 3a, d). The cold-side temperature was maintained at 25 °C, and the hot-side temperature was maintained at 60 °C. The characteristic peaks of $I_3^-$ and $I^-$ are at 150 cm$^{-1}$ (ref. 36) and 1050 cm$^{-1}$ (ref. 37), respectively. From the hot side to the cold side of the N-type i-TEC, the peak intensity of $I_3^-$ gradually decreases (Fig. 3b). The content of $I_3^-$ at the hot side is 3.5 times that at the cold side (Fig. 3c). This result indicates that the hydrophobic hot side attracts $I_3^-$ and the hydrophilic cold side repels $I_3^-$. Therefore, the content of $I_3^-$ gradually decreases from the hot side to the cold side. The peak intensity of $I_3^-$ gradually increases from the hot side to the cold side of the P-type i-TEC (Fig. 3e). The content of $I_3^-$ at the cold side is 6.2 times that at the hot side (Fig. 3f). Because the P-type i-TEC has the hydrophilic hot side and the hydrophobic cold side, it exhibits attraction to $I_3^-$ at the cold side and repulsion to $I_3^-$ at the hot side.

UV−vis absorption spectroscopy was used to measure the $I_3^-/I^-$ absorption capacity of hydrogels. The concentration change ratio of $I^-$ in the solution remains essentially unchanged after hydrogel soaking (Fig. S8). The concentration change ratio of $I_3^-$ in the solution after N-type hydrogel soaking increases rapidly at a temperature higher than 50 °C (Fig. S8 and Fig. 3g). The concentration change ratio of $I_3^-$ in the solution after P-type hydrogel soaking decreases quickly at a temperature higher than 50 °C (Fig. S8 and Fig. 3h). These results indicate that the interaction between the hydrophobic region of the hydrogel and $I_3^-$ is strong. Without a temperature gradient, there is no significant concentration difference of $I_3^-/I^-$ in N-type and P-type i-TECs (Fig. S9). The result indicates that the concentration difference of redox couples between the hot and cold sides should be caused by the hydrogel phase transition.

The $I_3^-$ and $I^-$ concentrations in the hydrogel i-TECs were also determined using energy dispersive X-ray spectra (EDS) mapping. The

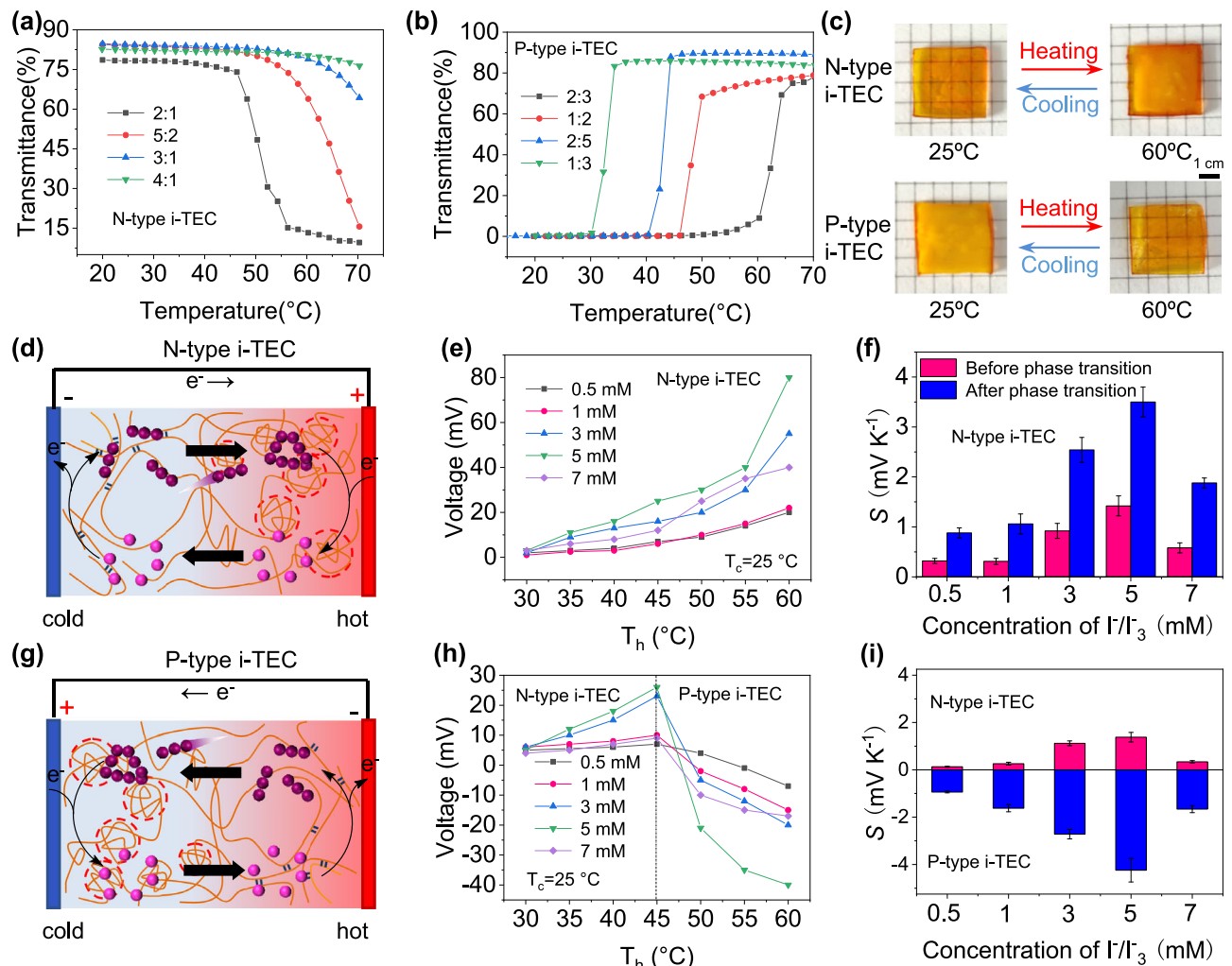

**Fig. 2 | Temperature-sensitive and thermoelectric properties of hydrogel i-TECs. a** Transmittance of the N-type i-TEC containing 5 mM $I_3^-$ with varying monomer mass ratios ($m_{MAA}$:$m_{DMAPS}$ = 2:1, 5:2, 3:1, 4:1). **b** Transmittance of the P-type i-TEC containing 5 mM $I_3^-$ with varying monomer mass ratios ($m_{MAA}$:$m_{DMAPS}$ = 2:3, 1:2, 2:5, 1:3). **c** Photos of hydrogel i-TECs at different temperatures. **d** Schematics of the N-type i-TEC (hot-side temperature ($T_h$) > phase transition temperature ($T_p$), cold-side temperature ($T_c$) < $T_p$). **e** Output voltage of the N-type i-TEC containing $x$ mM $I_3^-$ ($x$ = 0.5, 1, 3, 5, 7) with the increase of the hot-side temperature. The cold-side temperature is maintained at 25 °C. **f** Seebeck coefficient ($S$) of the N-type i-TEC. **g** Schematics of the P-type i-TEC ($T_h > T_p$, $T_c < T_p$). **h** Output voltage of the P-type i-TEC containing $x$ mM $I_3^-$ ($x$ = 0.5, 1, 3, 5, 7) with the increase of the hot-side temperature. The cold-side temperature is maintained at 25 °C. **i** Seebeck coefficient ($S$) of the P-type i-TEC. The error bars were calculated using the standard deviation of the measured Seebeck coefficient.

result shows that there is no significant difference in the $I^-$ distribution at the hot and cold sides of N-type and P-type i-TECs (Fig. S10). The content of $I_3^-$ in the N-type i-TEC is much larger at the hot side than at the cold side, and the amount of $I_3^-$ in the P-type i-TECs is the opposite (Fig. 3i and Fig. S10). These results indicate that the interaction between the hydrophobic region of the hydrogel and $I_3^-$ is stronger than with $I^-$.

**Performance optimization**

According to the reported work[38], KCl was added to the hydrogel i-TECs to increase the $S$. We tested the open-circuit voltage and short-circuit current of a single N-type i-TEC or P-type i-TEC at $\Delta T = 15$ K. Since the N–P conversion temperature of hydrogel i-TECs is close to 45 °C, the 45 °C cold-side temperature is more conducive to obtaining high $S$ than the 25 °C cold-side temperature (Fig. 4a) The optimized N-type and P-type i-TECs have the $S$ of 7.7 mV K$^{-1}$ and −6.3 mV K$^{-1}$, respectively (Figs. S11 and S12). The hydrogel i-TECs without KCl addition have low ionic conductivity ($\sigma$) of ~2 mS cm$^{-1}$ (Fig. S13). After adding KCl, the $\sigma$ of hydrogel i-TECs is about 25 mS cm$^{-1}$ (Fig. S14). KCl significantly reduces internal resistance of the hydrogel i-TECs. To

further explore the effect of KCl on the $S$ and short-circuit current of i-TECs, we tested cyclic voltammetry (CV) curve of i-TECs with/without KCl. The test temperature is 25 °C and the scanning rate is 50 mV s$^{-1}$. The i-TECs with KCl exhibit a smaller peak separation than the i-TECs without KCl (Fig. S15). This indicates that adding KCl can make i-TECs have faster electron transfer kinetics and better redox reversibility. In addition, the peak current intensity of redox reaction in the i-TECs with KCl is significantly higher than that in the i-TECs without KCl, indicating that adding KCl can increase the current output. The FTIR spectra of i-TECs with KCl show a decrease in the peaks of C=O stretching, C–O–C stretching and SO$_3^-$ groups (Fig. S16), indicating that K$^+$ ions interact with lone pair electrons at the oxygen (O) sites of the polymer chain side groups. The addition of KCl causes the C–O–C stretching toward longer wavelengths, further indicating the interaction between K$^+$ ions and O sites. Despite multiple cyclic establishing and removing temperature differences, the $S$ of i-TECs does not show significant changes (Fig. S17), demonstrating the repeatability and stability of the thermoelectric performance.

Water content will affect the thermoelectric performance of hydrogel i-TECs. We investigated the effect of water content on the $S$ of

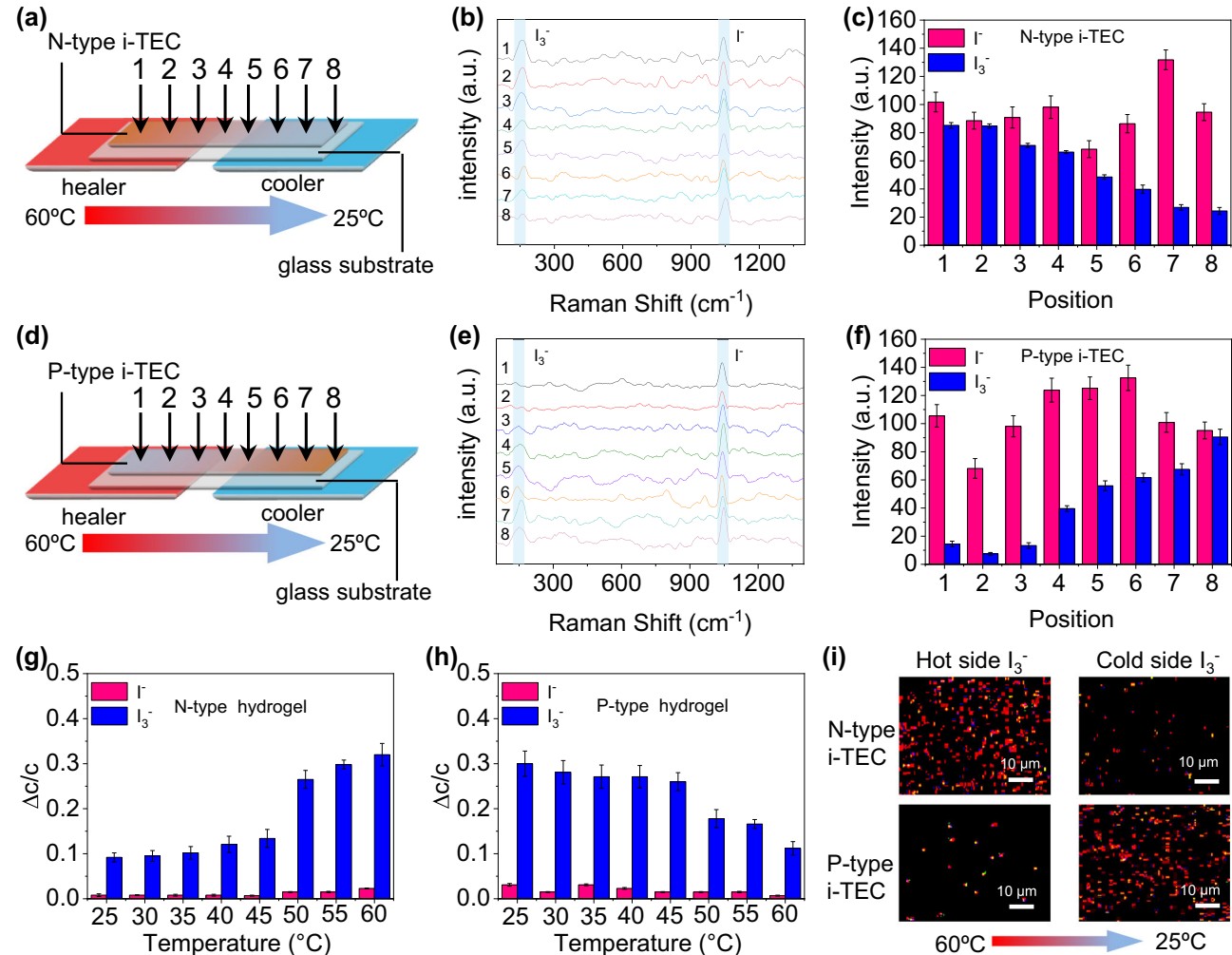

**Fig. 3 | Thermal gradient enhanced concentration difference of redox ions in hydrogel i-TEC. a** N-type i-TEC in-situ detection. **b** Raman spectra of the N-type i-TEC. **c** Intensity distribution of $I^-$ and $I_3^-$ in the N-type i-TEC. **d** P-type i-TEC in-situ detection. **e** Raman spectra of the P-type i-TEC. **f** Intensity distribution of $I^-$ and $I_3^-$ in the P-type i-TEC. **g**, **h** Concentration change ratio of $I^-$ and $I_3^-$ in the remaining solution after hydrogel soaking at different temperatures. **i** Energy dispersive X-ray spectra (EDS) mapping of $I_3^-$. The error bars were calculated using the standard deviation of the measured intensity and concentration change ratio.

i-TECs. With the continuous loss of water, the ion migration in the hydrogel will be blocked, leading to the decrease of Seebeck coefficient until the ions in the hydrogel can hardly migrate (Fig. S18a). To avoid the impact of water loss on the thermoelectric performance during the long-term test, we encapsulated the i-TECs with polyethylene film. The results show that water content and the $S$ of i-TECs remain basically unchanged within 5 days (Fig. S18b, c), demonstrating their long-term stability.

The N-type i-TEC and P-type i-TEC were coupled in a Π-shape (Fig. 4b). Compared with hydrogel i-TECs without KCl addition (Fig. S19), KCl addition significantly increases the output voltage, current and power of hydrogel i-TECs (Fig. 4c). Unless otherwise specified, 0.3 M KCl was added to the hydrogel i-TECs in the following experiments. The maximum power density ($P_{max}$) of a pair of N–P i-TECs is 10 μW, which is nearly equal to the sum of its two separate i-TECs (Fig. 4c). The results indicate the couple of N–P i-TECs is successful. When the output voltage is the sum of N-type and P-type i-TECs, and the current is close to either N-type i-TEC or P-type i-TEC, the matching resistance of N-type and P-type i-TECs will not cause excessive energy waste. The i-TECs with high $P_{max}/\Delta T^2$ are ideal for efficiently generating electricity from temperature differences. At a temperature difference of 15 K, the $P_{max}/\Delta T^2$ of N-type i-TEC is 0.76 mW m$^{-2}$ K$^{-2}$ and the $P_{max}/\Delta T^2$ of P-type i-TEC is 0.68 mW m$^{-2}$ K$^{-2}$ (Fig. 4d). We compared

the values of $S$, $P_{max}/\Delta T^2$ and $\eta_r$ with previously reported i-TECs (Fig. 4e, f)[17,20,26,38–41]. The results indicate that the comprehensive performance of our i-TECs exceeds that of the reported $I_3^-/I^-$-based i-TECs.

### Series integration and application

By connecting N-type and P-type i-TECs in series, high voltage and power can be obtained, making it a promising power source for microelectronic devices[42,43]. We tested the thermoelectric performance of multiple pairs of N–P i-TECs. The results show that the voltage and power generated by series integration of hydrogel i-TECs do not increase linearly (Fig. S20). Series connection increases voltage output while decreasing current due to increased internal resistance. As the number of N–P pairs increases to 10, the trade-off between voltage and current plateaus, resulting in a power output density of 85 μW. We constructed a flexible thermoelectric cell by connecting ten pairs of N–P i-TECs (Fig. 5a and Fig. S21) and a homemade Seebeck coefficient measurement device (Fig. S22). When the temperature difference is 35 K, the cell can generate 1.8 V voltage and 85 μW power (Fig. 5b–d). Compared with other series integrated hydrogel i-TECs, the thermoelectric device we prepared has high output voltage and power (Table S2). The rated voltage of some microelectronic devices is 1.5–3 V. Our cell can directly power some devices, such as electronic watches (Fig. 5e) and LED lights (Fig. 5f). As the total power consumption of

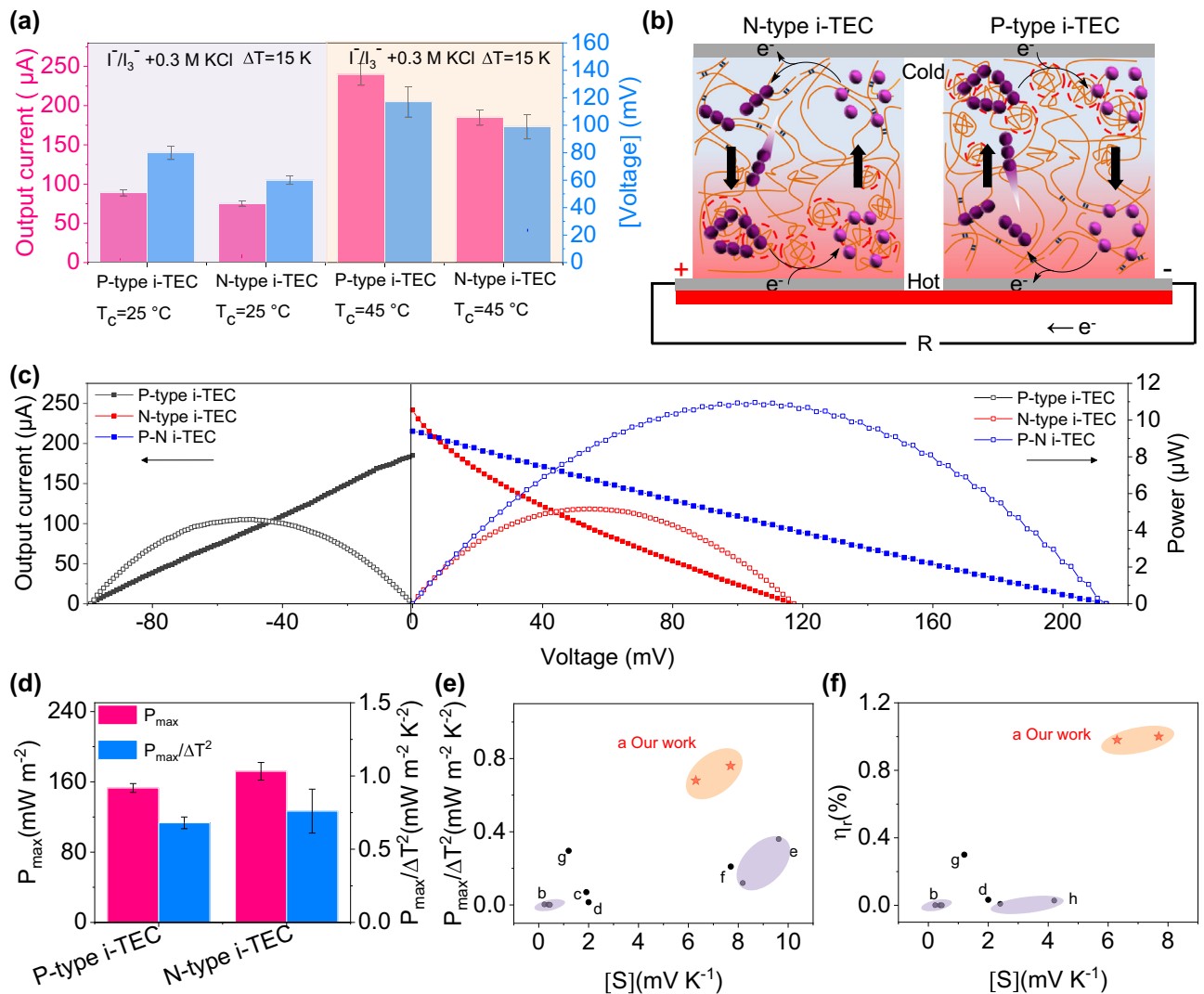

**Fig. 4 | Performance optimization. a** Output current and voltage of hydrogel i-TECs with KCl addition. **b** Couple of N-type and P-type i-TECs. **c** Voltage−current−power. **d** Maximum power density ($P_{max}$) and normalized instantaneous power density ($P_{max}/\Delta T^2$). **e** Comparison of the $S$ and $P_{max}/\Delta T^2$. The literature and data represented by letters (**a**–**h**) are listed in Table S1. **f** Comparison of the $S$ and Carnot-relative efficiency ($\eta_r$). The error bars were calculated using the standard deviation of the measured value.

electronics increases, the generated heat lowers operational performance and exacerbates device failure[44,45]. The central processing unit (CPU) generates a large amount of heat during normal operation, which will reduce the computer's stability and speed (Fig. 5g). The normal working temperature of CPU is 60 °C (Fig. 5h). Lowering the temperature will improve its efficiency and lifespan[46]. Hydrogels can be used to cool electronics and even generate electricity[47]. We placed a hydrogel of 30 mm × 30 mm × 3 mm on the CPU surface (Fig. 5i), resulting in a 15.1 K temperature drop from 76.2 °C to 61.1 °C (Fig. 5j). In addition, four pairs of N−P i-TECs in series can generate 0.45 V voltage and 20 µW power when covered on the CPU surface of normally operating computer (Fig. S23). This confirms the potential application of our cells in cooling devices and power generation.

## Discussion
In this work, we report $I_3^-/I^-$-based series integrated hydrogel i-TECs for low-grade heat harvesting. The N−P conversion of i-TECs is realized by the hydrogel phase transition. The concentration difference of redox ions is enhanced due to the strong interaction between the hydrophobic region of the hydrogel and $I_3^-$, thus increasing the $S$. The optimized $S$ for the N-type i-TEC is 7.7 mV K$^{-1}$, the P-type i-TEC is

−6.3 mV K$^{-1}$. By connecting ten pairs of N−P i-TECs in series, our cell shows high voltage (1.8 V) and output power (85 µW) that can power the electronic watch and LED light. This work expands the possibility of low-grade heat harvesting, especially with hydrogel i-TECs.

## Methods
### Materials
MAA (99.0%), DMAPS (97.0%), iodine ($I_2$) (99.5%), potassium iodide (KI) (99.8%) and 2-hydroxy-2-methylpropiophenone (photoinitiator) (97.0%) were purchased from Aladdin Biochemical Technology Co., Ltd. (Shanghai, China). Graphite paper with a thickness of 200 µm was purchased from Jinglong Carbon Technology Co., Ltd. (Beijing, China). Deionized water with a resistivity of 18.2 MΩ cm was used throughout the experiments.

### Preparation of hydrogel i-TECs
The hydrogel was prepared by one-pot copolymerization of MAA and DMAPS monomers. Typically, MAA and DMAPS were dissolved in deionized water. The photoinitiator was added to the solution. The total monomer concentration was 30 wt%. The photoinitiator concentration was 0.2 wt%. The mixture was injected into a sandwich

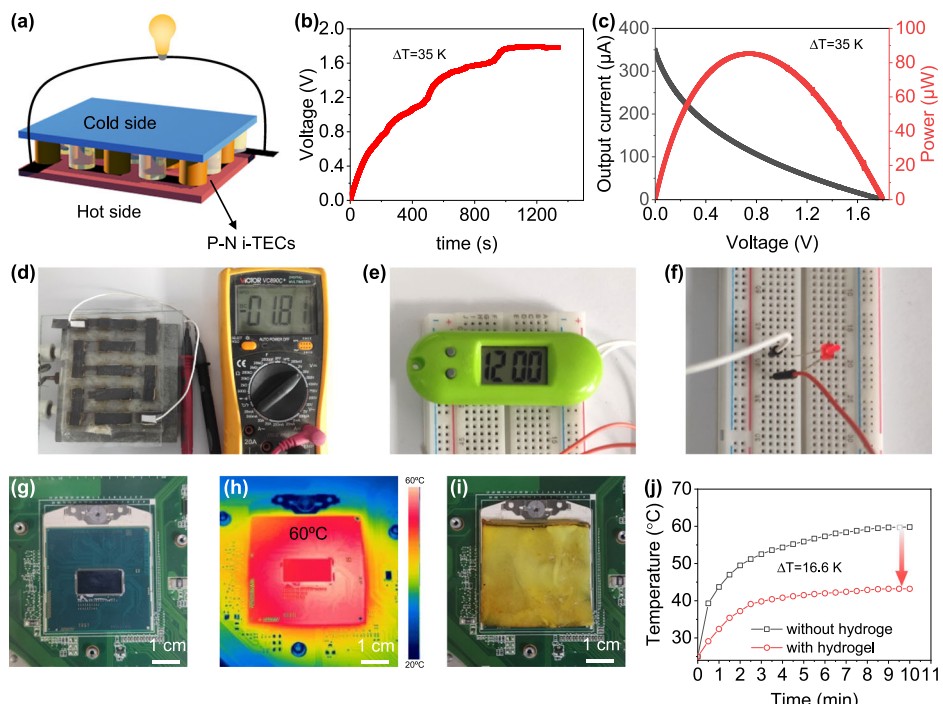

**Fig. 5 | Series integrated hydrogel i-TECs. a** Series integration. **b** Output voltage of ten pairs of N–P i-TECs. **c** Output current and power of ten pairs of N–P i-TECs. **d** Photo of the flexible thermoelectric cells manufactured by ten pairs of N–P i-TECs. The cold-side temperature is 25 °C, and the hot-side temperature is 60 °C. **e** Power the electronic watch. **f** Power the LED light. **g** Central processing unit (CPU) photo. **h** Surface temperature of the CPU under normal operation. **i** Hydrogel i-TEC covering the CPU surface to cool CPU and generate electricity. **j** Temperature change of CPU surface before and after hydrogel covering.

module with a certain size and irradiated with ultraviolet light for 3 h to produce the hydrogel. The power of the ultraviolet lamp is 50 W, and the height of the sample from the ultraviolet lamp is 20 cm. To eliminate residual chemicals, the hydrogel was rinsed with cold water (about 5 °C). The hydrogel prepared with the feed mass ratio of MAA and DMAPS of 2:1 was denoted as the N-type hydrogel. The hydrogel prepared with the feed mass ratio of MAA and DMAPS of 1:2 was denoted as the P-type hydrogel.

The $I_3^-/I^-$ solution was prepared by dissolving KI and $I_2$ (molar ratio 2:1) in deionized water. The power of the ultrasonic cleaner is 20 W, and the ultrasound lasts for 8 h to completely dissolve KI and $I_2$. After the reaction of $I^-$ and $I_2$ to form $I_3^-$, the molar ratio of $I^-$ and $I_3^-$ was 1:1. The hydrogel i-TEC was prepared by immersing the hydrogel in the $I_3^-/I^-$ solution for 1 h. The residual solution was removed from the hydrogel surface by air flow.

### Preparation of series integrated hydrogel i-TECs
To generate spacing between the N-type and P-type i-TECs, 5 mm thick PDMS was hollowed out at equal distances as a cushion. The N-type (5 mm × φ 10 mm) and P-type (5 mm × φ 10 mm) i-TECs were carefully inserted into the hollows. Graphite paper was used to connect the i-TECs. Finally, the i-TECs were packaged using polypropylene film.

### Characterization
The hydrogel turbidity was characterized using Shimadzu UV-2600 spectrophotometer, and the sample was heated with an automatic temperature control accessory. UV–vis absorption spectra were recorded using Shimadzu UV-2600 spectrophotometer to determine the relative concentration change of $I_3^-/I^-$. EDS were recorded using Hitachi SU8010 scanning electron microscope. Raman spectra were recorded on the i-Raman plus 785S spectrometer equipped with a Xe lamp with a 532 nm laser excitation (maximum laser power: 40 mW, minimum laser spot diameter: 85 μm, spectral resolution: 4.5 cm⁻¹@614 nm, working temperature: 25 °C).

### Mechanical properties
The uniaxial tensile and compression tests were performed at room temperature using electronic universal testing equipment (E43.104, MTS). The tensile test was conducted using a rectangular hydrogel (10 mm × 30 mm × 3 mm) at a tensile speed of 50 mm min⁻¹. The compression test was conducted using a cylindrical hydrogel (10 mm × φ 10 mm) at a compression speed of 2 mm min⁻¹. The toughness was estimated from the stress–strain curve. The dissipated energy was determined by the region between the loading and unloading curves.

### Conductivity
The ionic conductivity at room temperature was measured using an electrochemical workstation (CHI760D). A hydrogel sheet (5 mm × 10 mm × 3 mm) was layered between the two electrodes. The scanning frequency is from 0.05 to $10^6$ Hz. The hydrogel resistance was determined by calculating the first intercept of high frequency on the horizontal axis from Nyquist plots. The ionic conductivity can be calculated by the following equation:

$$\sigma = \frac{L}{AR} \tag{1}$$

where $\sigma$ (mS cm⁻¹) is the ionic conductivity, $L$ (cm) is the distance between the two electrodes, $A$ (cm²) is the cross-sectional area and $R$ (Ω) is the equivalent series resistance.

### Thermoelectric properties
The temperature on both sides of the i-TECs was adjusted by the Peltier device to form a temperature difference. Thermocouple was used for real-time temperature monitoring. A CHI760D electrochemical workstation was used to detect the voltage and current generated by the i-TECs. The Seebeck coefficient ($S$) was calculated by the following

equation:

$$S = \frac{V_h - V_c}{T_h - T_c} \quad (2)$$

where $S$ (mV K$^{-1}$) is the Seebeck coefficient, $V_h$ (mV) is the hot-side voltage, $V_c$ (mV) is the cold-side voltage, $T_h$ (K) is the hot-side temperature and $T_c$ (K) is the cold-side temperature.

The Carnot-relative efficiency $\eta_r$ is defined as:

$$\eta_r = \frac{\eta}{\eta_c} \quad (3)$$

where $\eta$ is the conversion efficiency, $\eta_c$ is the Carnot efficiency which is the limiting efficiency of a heat engine. The conversion efficiency ($\eta$) was calculated by the following equation:

$$\eta = \frac{P_{max}}{P_{heat}} = \frac{P_{max}}{\kappa_{eff} \times A \times (\Delta T/d)} \quad (4)$$

where $\kappa_{eff}$ (W m$^{-1}$ K$^{-1}$) is the effective thermal conductivity, $A$ (m$^2$) is the cross-sectional area, $\Delta T$ (K) is the temperature difference and $d$ (m) is the distance between the two electrodes.

The Carnot efficiency ($\eta_c$) was defined as:

$$\eta_c = \frac{\Delta T}{T_h} \quad (5)$$

where $T_h$ (K) is the hot-side temperature.

Thus, $\eta_r$ can be calculated by the following equation:

$$\eta_r = \frac{P_{max}/(\kappa_{eff} \times A \times (\Delta T/d))}{\Delta T/T_h} \quad (6)$$

## Reporting summary

Further information on research design is available in the Nature Portfolio Reporting Summary linked to this article.

## Data availability

All data supporting the findings of this study are available within the article and its Supplementary files. Any additional requests for information can be directed to, and will be fulfilled by, the corresponding authors. Source data are provided with this paper.

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

## Acknowledgements

This work is supported by the National Natural Science Foundation of China (22474132, X.Z., 22090050, F.X.) and National Key R&D Program of China (2021YFA1200403, F.X.).

## Author contributions

X.Z. and F.X. proposed the research direction and guided the project. J.S. designed and performed the experiments. J.S., X.H., Y.D., X.Z., and F.X. analyzed and discussed the results. All authors wrote the manuscript.

## Competing interests

The authors declare no competing interests.
