## [Peer Review file · Nature Communications]

N-type and P-type series integrated hydrogel thermoelectric cells for low-grade heat harvesting

Corresponding Author: Professor Fan Xia

Version 1:

Reviewer comments:

Reviewer #1

(Remarks to the Author)

This manuscript by Shen et al, described a novel technique of transforming low-grade heat to electric energy. The main innovation here is to offer a hydrogel phase transition strategy for preparing P/N hydrogel i-TECs with same redox couple, and demonstrate series integration of hydrogel i-TECs for low-grade heat harvesting. According to the previous studies, the academia is fully aware of the importance of the effective serial integration of P-N i-TECs. Thus, the development of corresponding technique is critical as well.

This is a very interesting manuscript and the results shown in this manuscript are quite convincing. A large number of results are displayed in the main text and the SI. The manuscript is well written and the story line is fluent and clear to the reader, very convincing and eye-catching definitely. I strongly recommend publishing it in Nature Communications. However, regarding this manuscript, there are still some minor issues and comments which need to be resolved prior to the final acceptance.

1. The introduction mentions challenges related to the limited Seebeck coefficient (Se) and series integration but could be strengthened by providing more detail on these challenges. Outlining the typical range of Se values in conventional i-TECs, emphasizing the limitations of current series integration strategies, such as cross-contamination, as well as highlighting the difference from the previously reported I-/I3- i-TECs would add significant value to the discussion.
2. From the available data (Figure 3g, h), the ratio of I-/I3- redox couple in this gel is not 1:1. Does this have an effect on the Se? It would be helpful to include a control experiment in the Raman result that does not involve a temperature gradient. This would help to confirm that the observed concentration differences are indeed driven by the thermal gradient and not by other factors, thus strengthening the validity of the findings.
3. As for the performance optimization, the addition of KCl makes the Se and short-circuit current of the TECs significantly improved. There is little evidence on the mechanism of Se enhancement. Although relevant literature is quoted (Sci. Adv. 8, eabl5318 (2022)), both of these materials have the characteristics of temperature-sensitive phase transition. However, there are differences in chemical structure. Where does the K+ bind to the polymer in this study? What is the form of the salt-induced complexation?
4. Understand that the samples were tested in ambient environment. Will there be water loss during long-term measurements and how does this affect thermoelectric performance?
5. Can the phase transition process of P-N i-TECs be repeated and how does it affect their thermoelectric performance?
6. Is the voltage and power produced by series integration of P-N i-TECs linearly increasing? If not, what are the optimum values?
7. A few recent works on i-TECs can provide a broad context, e.g. 10.1016/j.decarb.2023.100003; 10.1039/D1TA10508F; 10.1002/adfm.202214563; 10.1007/s40820-023-01077-7
8. On line 171 of page 8, "the hot cold-side" should be "the hot side".

9. In Figure 4 e and f, the meaning of the letters (a-e) is unclear.

Reviewer #2

(Remarks to the Author)

In this work, the author proposes a phase transition-based ionic thermoelectric cell suitable for near-room-temperature applications. The experimental data is robust, and the measurement techniques are standard. Ionic thermoelectric materials present challenges due to charge accumulation, leading to carrier concentration inhomogeneity and inconsistent output performance over time. Addressing degradation and understanding the underlying mechanisms are crucial. The author demonstrates significant output performance from n- and p-type thermoelectric cells, providing a foundation for future research. Certain technical questions and established phenomena from bulk ionic character inorganic thermoelectric research may be applicable to this study. The manuscript warrants consideration for scientific publication.

Comments:

1. The introduction inaccurately characterizes industrial heat as exclusively low-grade (5-20 K). Industries such as chemical, automotive, nuclear, and thermal power generation produce significant amounts of high-temperature heat. As the manuscript focuses on low-grade heat, the author should provide a more realistic assessment of industrial heat sources. Additionally, thermomagnetic technology, alongside thermoelectric (Bi-Te, Pb-Te) solutions, should be considered as potential alternatives for low-grade energy harvesting. Author may take a look on the recent literature for better discussion of intro part. (1) Constructive approach towards design of high-performance thermoelectrics, thermal diodes, and thermomagnetic devices for energy generation applications. In *Energy Harvesting and Storage Devices* (pp. 80-107) 2023. CRC Press. (2) *ACS Appl. Mater. Interfaces* 2023, 15, 29, 35140–35148
2. Fig 1c, How negative Seebeck coefficients are possible for p-type ion, I assumed its positive ions or it is something else, please clarify it?
3. The author attributes the enhancement of Seebeck coefficients, crucial for achieving a high power factor, to a phase transition. However, the underlying scientific mechanism remains unclear. While the coexistence of two phases at the phase boundary is suggested as a potential reason for high Seebeck coefficients in inorganic ionic thermoelectric materials, the author should correlate this explanation with the behavior of the present materials and provide a comparative analysis. See example: (1) *J. Electron. Mater.* 49, 2855–2861 (2020) (2) *J. Phys. D: Appl. Phys.* 54 115503 (2021)
4. Are starting chemical purity, reaction temperature, and reaction time crucial factors influencing the obtained properties? If so, detailed experimental information should be provided to ensure reproducibility in future work.
5. The experimental setup shown in Figure S6 was used for measuring Seebeck coefficients. Was this measurement performed in a vacuum or inert gas atmosphere? Were standard samples, such as constantan or nickel metal, used to verify the measurement reliability?
6. The output measurement of the device prepared by coupling n- and p-type materials is unclear. The author should include a schematic or actual image of the device, along with details of any standard instruments used. While the extraordinary output performance with such a small temperature gradient is impressive, a comparison with the output power of flexible inorganic/organic materials tested under similar conditions for low-grade waste heat recovery would strengthen the findings. The author could incorporate this comparison using a bar diagram or table, referencing relevant studies. (1) *ACS Appl. Mater. Interfaces* 2022, 14, 34, 38642–38650 (2) *ACS Appl. Mater. Interfaces* 2023, 15, 40, 46962–46970

Version 2:

Reviewer comments:

Reviewer #1

(Remarks to the Author)

Reviewer #2

(Remarks to the Author)

Author have revised manuscript. I have no further questions.

Point-by-Point Response to the Reviewer 1's comments

Note: The italic font is the comment. The regular font is our response. The highlighted sentences are the modified text included in the manuscript.

This manuscript by Shen et al, described a novel technique of transforming low-grade heat to electric energy. The main innovation here is to offer a hydrogel phase transition strategy for preparing P/N hydrogel i-TECs with same redox couple, and demonstrate series integration of hydrogel i-TECs for low-grade heat harvesting. According to the previous studies, the academia is fully aware of the importance of the effective serial integration of P-N i-TECs. Thus, the development of corresponding technique is critical as well.

This is a very interesting manuscript and the results shown in this manuscript are quite convincing. A large number of results are displayed in the main text and the SI. The manuscript is well written and the story line is fluent and clear to the reader, very convincing and eye-catching definitely. I strongly recommend publishing it in Nature Communications. However, regarding this manuscript, there are still some minor issues and comments which need to be resolved prior to the final acceptance.

Response: We are grateful to the reviewer for his/her positive and encouraging feedback. We also sincerely appreciate the reviewer for his/her constructive comments, which will help improve the quality of our manuscript. According to these comments, we have carefully revised our manuscript. The point-by-point response is as follows.

- 1. The introduction mentions challenges related to the limited Seebeck coefficient (Se) and series integration but could be strengthened by providing more detail on these challenges. Outlining the typical range of Se values in conventional i-TECs, emphasizing the limitations of current series integration strategies, such as cross-contamination, as well as highlighting the difference from the previously reported I/I_3^- i-TECs would add significant value to the discussion.*

Response: We sincerely appreciate the reviewer for this insightful comment. In

response to this comment, we added corresponding detailed discussions in the introduction section to further explain the challenges and the difference from the previously reported i-TECs.

Despite the progress made, the i-TECs still face challenges such as limited S and required series integration. For example, the absolute S of I_3^-/I^- , $Fe(CN)_6^{-3}/Fe(CN)_6^{-4}$ and Fe^{3+}/Fe^{2+} i-TECs is 0.5-0.8 mV K⁻¹, 1.4 mV K⁻¹ and 1.04 mV K⁻¹, respectively.

The P-type and N-type i-TECs are typically different redox couples, which may cause cross-infection. For example, $Fe(CN)_6^{-3}/Fe(CN)_6^{-4}$ exhibits good stability under neutral and alkaline conditions, but produces highly toxic hydrogen cyanide under acidic conditions, making it incompatible with acidic i-TECs such as Fe^{3+}/Fe^{2+} .

Here, we report for the first time the N-P conversion of hydrogel i-TECs with same redox couple, and demonstrate series integration of hydrogel i-TECs for low-grade heat harvesting (Figure 1a). Different from the previously reported i-TECs, the concentration difference of redox couples in our i-TECs is caused by the hydrogel phase transition, which does not depend on the carrier.

2. *From the available data (Figure 3g, h), the ratio of I^-/I_3^- redox couple in this gel is not 1:1. Does this have an effect on the S_e ? It would be helpful to include a control experiment in the Raman result that does not involve a temperature gradient. This would help to confirm that the observed concentration differences are indeed driven by the thermal gradient and not by other factors, thus strengthening the validity of the findings.*

Response: Thanks for your helpful suggestion. We fully agree that the Raman result that does not involve a temperature gradient is important for confirming the observed concentration differences. Therefore, we performed the control experiment. Without a temperature gradient, there is no significant concentration difference of I_3^-/I^- in N-type and P-type i-TECs (Figure S9). The result indicates that the concentration difference of redox couples between the hot and cold sides should be caused by the hydrogel phase transition.

In response to this comment, we added Figure S9 and two sentences in the revised

manuscript.

Figure S9. (a) N-type i-TEC in-situ detection. (b) Raman spectra of the N-type i-TEC. (c) Intensity distribution of I_3^-/Γ in the N-type i-TEC. (d) P-type i-TEC in-situ detection. (e) Raman spectra of the P-type i-TEC. (f) Intensity distribution of I_3^-/Γ in the P-type i-TEC.

3. As for the performance optimization, the addition of KCl makes the Se and short-circuit current of the TECs significantly improved. There is little evidence on the mechanism of Se enhancement. Although relevant literature is quoted (*Sci. Adv.* 8, eabl5318 (2022)), both of these materials have the characteristics of temperature-sensitive phase transition. However, there are differences in chemical structure. Where does the K^+ bind to the polymer in this study? What is the form of the salt-induced complexation?

Response: We sincerely thank the reviewer for this insightful comment. To further explore the effect of KCl on the S and short-circuit current of i-TECs, we tested cyclic voltammetry (CV) curve of i-TECs with/without KCl. The test temperature is 25 °C and the scanning rate is 50 mV s⁻¹. The i-TECs with KCl exhibit a smaller peak separation than the i-TECs without KCl (Figure S15). This indicates that adding KCl can make i-TECs have faster electron transfer kinetics and better redox reversibility. In addition, the peak current intensity of redox reaction in the i-TECs with KCl is

significantly higher than that in the i-TECs without KCl, indicating that adding KCl can increase the current output. The FTIR spectra of i-TECs with KCl show a decrease in the peaks of C=O stretching, C-O-C stretching and SO₃⁻ groups (Figure S16), indicating that K⁺ ions interact with lone pair electrons at the oxygen (O) sites of the polymer chain side groups. The addition of KCl causes the C-O-C stretching towards longer wavelengths, further indicating the interaction between K⁺ ions and O sites.

In response to this comment, we added Figure S15, Figure S16 and a paragraph in the revised manuscript.

Figure S15. (a) CV curve of the N-type i-TEC with/without KCl at 25°C. (b) CV curve of the P-type i-TEC with/without KCl at 25°C.

Figure S16. (a) FTIR spectra of the N-type i-TEC with/without KCl. (b) FTIR spectra of the P-type i-TEC with/without KCl.

4. Understand that the samples were tested in ambient environment. Will there be water loss during long-term measurements and how does this affect thermoelectric performance?

Response: We sincerely appreciate the reviewer for this helpful comment. Water

content will affect the thermoelectric performance of hydrogel i-TECs. We investigated the effect of water content on the S of i-TECs. With the continuous loss of water, the ion migration in the hydrogel will be blocked, leading to the decrease of Seebeck coefficient until the ions in the hydrogel can hardly migrate (Figure S18a). To avoid the impact of water loss on the thermoelectric performance during the long-term test, we encapsulated the i-TECs with polyethylene film. The results show that water content and the S of i-TECs remain basically unchanged within five days (Figure S18b and Figure S18c), demonstrating their long-term stability.

In response to this comment, we added Figure S18 and some sentences in the revised manuscript.

Figure S18. (a) Effect of water content on the S of i-TECs. (b) Change in water content and the S of the N-type i-TECs encapsulated with polyethylene film. (c) Change in water content and the S of the P-type i-TECs encapsulated with polyethylene film.

5. *Can the phase transition process of P-N i-TECs be repeated and how does it affect their thermoelectric performance?*

Response: This is a good question. We conducted cyclic test on the transmittance of i-TECs at 25°C and 60°C . The i-TECs exhibit rapid reversible changes in transmittance (Figure S6), implying their exceptional stability and reproducibility. Despite multiple cyclic establishing and removing temperature differences, the S of i-TECs does not show significant changes (Figure S17), demonstrating the repeatability and stability of the thermoelectric performance.

In response to this comment, we added Figure S6, Figure S17 and some sentences in

the revised manuscript.

Figure S6. Transmittance of i-TECs under cyclic heating and cooling at 25°C and 60°C. (a) N-type i-TEC. (b) P-type i-TEC.

Figure S17. Seebeck coefficient of i-TECs under cyclic establishing and removing temperature differences.

6. *Is the voltage and power produced by series integration of P-N i-TECs linearly increasing? If not, what are the optimum values?*

Response: We sincerely appreciate the reviewer for this insightful comment. We tested the thermoelectric performance of multiple pairs of N-P i-TECs. The results show that the voltage and power generated by series integration of hydrogel i-TECs do not increase linearly (Figure S20). Series connection increases voltage output while decreasing current due to increased internal resistance. As the number of N-P pairs increases to 10, the trade-off between voltage and current plateaus, resulting in a power output density of 85 μW .

In response to this comment, we added Figure S20 and some sentences in the revised manuscript.

Figure S20. Thermoelectric performance of multiple pairs of N-P i-TECs at $\Delta T = 35$

K.

7. A few recent works on i-TECs can provide a broad context, e.g. [10.1016/j.decarb.2023.100003](https://doi.org/10.1016/j.decarb.2023.100003); [10.1039/D1TA10508F](https://doi.org/10.1039/D1TA10508F); [10.1002/adfm.202214563](https://doi.org/10.1002/adfm.202214563); [10.1007/s40820-023-01077-7](https://doi.org/10.1007/s40820-023-01077-7)

Response: We sincerely appreciate the reviewer for this constructive suggestion to strengthen the manuscript. We carefully studied these literatures and added the citations in the revised manuscript.

- 11 He, Q. J., Cheng, H. L. & Ouyang, J. Y. Flexible combinatorial ionic/electronic thermoelectric converters to efficiently harvest heat from both temperature gradient and temperature fluctuation. *DeCarbon* **1**, 100003 (2023).
- 12 Zhou, Y., Yao, C. L., Lin, X. X., Oh, J., Tian, J. J., Yang, W. Z., He, Y. J., Ma, Y. H., Yang, K., Ai, B., Sun, K., Fu, Z. P., Lu, Y. L., Li, F., Yang, C. D. & Chen, S. S. Ion exchange induced efficient N-type thermoelectrics in solid-state. *Adv. Funct. Mater.* **33**, 2214563 (2023).
- 13 He, Y. J., Li, S. W., Chen, R., Liu, X., Odunmbaku, G. O., Fang, W., Lin, X. X., Ou, Z. P., Gou, Q. Z., Wang, J. C., Ouedraogo, N. A. N., Li, J., Li, M., Li, C., Zheng, Y. J., Chen, S. S., Zhou, Y. L. & Sun, K. Ion-electron coupling enables ionic thermoelectric material with new operation mode and high energy density. *Nano-Micro Lett.* **15**, 101 (2023).
- 14 Liu, Y., Zhang, Q., Odunmbaku, G. O., He, Y. J., Zheng, Y. J., Chen, S. S., Zhou, Y. L., Li, J., Li, M. & Sun, K. Solvent effect on the Seebeck coefficient of $\text{Fe}^{2+}/\text{Fe}^{3+}$ hydrogel thermogalvanic cells. *J. Mater. Chem. A* **10**, 19690-19698 (2022).

8. On line 171 of page 8, “the hot cold-side” should be “the hot side”.

Response: We sincerely appreciate the reviewer for this mistake. We corrected it in the revised manuscript.

The cold-side temperature was maintained at 25°C , and the hot-side temperature was maintained at 60°C .

9. In Figure 4 e and f, the meaning of the letters (a-e) is unclear.

Response: We sincerely appreciate the reviewer for this constructive suggestion to strengthen the manuscript. We added the explanation for the letters in Figure 4e and f in the supplementary information (Table S1).

Figure 4. (e) Comparison of the S and $P_{\max}/\Delta T^2$. The literature and data represented by letters (a-h) are listed in Table S1. (f) Comparison of the S and Carnot-relative efficiency (η_r).

Table S1. Comparison of thermoelectric performance between our i-TECs and the reported I_3^-/I^- -based i-TECs

Letter	Electrolyte	S (mV K ⁻¹)	$P_{\max}/\Delta T^2$ (mW m ⁻² K ⁻²)	η_r (%)	Ref.
a	Phase transition hydrogel/KCl	7.7	0.76	1	This work
	[C ₂ mim] [BF ₄] ionic liquid	-6.3	0.68	0.98	
b	PNIPAM nanogel/aqueous solution	-0.23	0.0029	0.00075	1
		-0.39	0.0017	0.00025	
c	KCl aqueous solution	1.91	0.07	--	2
d	Methyl-cellulose/KCl aqueous solution	-2	0.0147	0.033	3
e	Dimethyl carbonate/ethylene carbonate solvent	-8.18	0.12	--	4
		9.62	0.36	--	
f	CsCl aqueous solution	7.7	0.21	--	5
g	α -CD/KCl aqueous solution	1.2	0.298	0.3	6
h	solution	2.4	--	0.009	7
		4.2	--	0.028	

Point-by-Point Response to the Reviewer 2's comments

Note: The italic font is the comment. The regular font is our response. The highlighted sentences are the modified text included in the manuscript.

In this work, the author proposes a phase transition-based ionic thermoelectric cell suitable for near-room-temperature applications. The experimental data is robust, and the measurement techniques are standard. Ionic thermoelectric materials present challenges due to charge accumulation, leading to carrier concentration inhomogeneity and inconsistent output performance over time. Addressing degradation and understanding the underlying mechanisms are crucial. The author demonstrates significant output performance from n- and p-type thermoelectric cells, providing a foundation for future research. Certain technical questions and established phenomena from bulk ionic character inorganic thermoelectric research may be applicable to this study. The manuscript warrants consideration for scientific publication.

Response: We are grateful to the reviewer for his/her positive and encouraging feedback. We also sincerely appreciate the reviewer for his/her constructive comments, which will help improve the quality of our manuscript. According to these comments, we have carefully revised our manuscript. The point-by-point response is as follows.

- 1. The introduction inaccurately characterizes industrial heat as exclusively low-grade (5-20 K). Industries such as chemical, automotive, nuclear, and thermal power generation produce significant amounts of high-temperature heat. As the manuscript focuses on low-grade heat, the author should provide a more realistic assessment of industrial heat sources. Additionally, thermomagnetic technology, alongside thermoelectric (Bi-Te, Pb-Te) solutions, should be considered as potential alternatives for low-grade energy harvesting. Author may take a look on the recent literature for better discussion of intro part. (1) Constructive approach towards design of high-performance thermoelectrics,*

thermal diodes, and thermomagnetic devices for energy generation applications. In Energy Harvesting and Storage Devices (pp. 80-107) 2023. CRC Press. (2) ACS Appl. Mater. Interfaces 2023, 15, 29, 35140–35148

Response: We sincerely appreciate the reviewer for this constructive suggestion to strengthen the manuscript. In response to this comment, we revised the manuscript to make its expression more accurate.

Low-grade heat, such as solar heat, low-temperature industrial waste heat and human body heat, is widely present in nature but is generally discarded.

In recent years, thermoelectric conversion technologies, such as organic Rankine cycle, Kalina cycle, thermomagnetic effect and thermoelectric effect, have been developed to directly convert heat into electricity.

7 Singh, S., Liu, N., Zhang, Y., Nozariasbmarz, A., Karan, S. K., Raman, L., Goyal, G. K., Sharma, S., Li, W. J., Priya, S. & Poudel, B. High-performance thermomagnetic Gd-Si-Ge alloys. *ACS Appl. Mater. Interfaces* **15**, 35140-35148 (2023).

2. *Fig 1c, how negative Seebeck coefficients are possible for p-type ion, I assumed its positive ions or it is something else, please clarify it?*

Response: We sincerely appreciate the reviewer for this insightful comment. In response to this comment, we added a supplementary note to explain the N-type and P-type conversion of hydrogel i-TECs in the supplementary information.

Supplementary Note

The Seebeck coefficient (S) of thermocells is defined as:

(1)

where, ΔE is the open-circuit voltage, n is the number of electrons in the redox reaction, F is the Faraday constant and ΔS is the entropy difference.

Based on the Nernst equation, if a redox reaction $A + ne \leftrightarrow B$ reaches equilibrium, the equilibrium potential (E) can be expressed as:

(2)

where, E^0 is the standard potential, R is the gas constant, α_A and α_B are the

activities of the oxidation and reduction species, respectively. According to the equation $a = c \cdot \gamma$, where γ is the activity coefficient and c is the concentration, equation (2) can be expressed as:

$$\text{[Redacted Equation (3)]} \quad (3)$$

where, E_f is the formal potential.

$$\text{[Redacted Equation (4)]} \quad (4)$$

According to equation (1), S can be expressed as:

$$\text{[Redacted Equation (5)]} \quad (5)$$

where, E_H and E_C are the potentials at the hot and cold electrodes. By combining equations (3) and (5), S can be specified as:

$$\text{[Redacted Equation (6)]} \quad (6)$$

For the pristine I_3^-/I^- ($I_3^- + 2e^- \rightarrow 3I^-$), the concentrations of redox species at the two sides are equal, namely, $[I_3^-]_h = [I_3^-]_c$, $[I^-]_h = [I^-]_c$ (subscript 'h' represents the hot side, 'c' represents the cold side). According to equation (6), the Seebeck coefficient (S) is expressed as:

$$\text{[Redacted Equation (7)]} \quad (7)$$

where, ΔS_{re} is the partial molar entropy.

Generally, the S of the pristine I_3^-/I^- is positive, equivalent to N-type.

When $[I_3^-]_h \neq [I_3^-]_c$, the corresponding S is calculated by equation (6) as follows:

$$\text{[Redacted Equation (8)]} \quad (8)$$

If $[I_3^-]_h < [I_3^-]_c$, the second term on the right side of equation (8) will result in a negative concentration entropy. Thus, it is possible to reverse S from a positive value to a negative value.

When the temperature exceeds the phase transition temperature (T_p), the hydrogel

becomes hydrophilic. Under a certain temperature difference (hot-side temperature (T_h) > T_p , cold-side temperature (T_c) < T_p), the hot side is hydrophilic (repelling I_3^-) and the cold side is hydrophobic (attracting I_3^-). The concentration of I_3^- at the hot side is lower than that at the cold side. When $[I_3^-]_h < [I_3^-]_c$, the S is a positive plus negative value. Therefore, the S may be negative, equivalent to P-type.

3. *The author attributes the enhancement of Seebeck coefficients, crucial for achieving a high power factor, to a phase transition. However, the underlying scientific mechanism remains unclear. While the coexistence of two phases at the phase boundary is suggested as a potential reason for high Seebeck coefficients in inorganic ionic thermoelectric materials, the author should correlate this explanation with the behavior of the present materials and provide a comparative analysis. See example: (1) J. Electron. Mater. 49, 2855–2861 (2020) (2) J. Phys. D: Appl. Phys. 54 115503 (2021)*

Response: We sincerely appreciate the reviewer for this insightful comment. In response to this comment, we added a supplementary note to explain the increase in Seebeck coefficients hydrogel i-TECs in the supplementary information.

Supplementary Note, please refer to the previous response.

4. *Are starting chemical purity, reaction temperature, and reaction time crucial factors influencing the obtained properties? If so, detailed experimental information should be provided to ensure reproducibility in future work.*

Response: We sincerely appreciate the reviewer for this constructive suggestion to strengthen the manuscript. In response to this suggestion, we added the experiment information in the revised manuscript.

Methacrylic acid (MAA) (99.0%), 3-dimethyl(methacryloyloxyethyl)ammonium propanesulfonate (DMAPS) (97.0%), iodine (I_2) (99.5%), potassium iodide (KI) (99.8%) and 2-hydroxy-2-methylpropiophenone (photoinitiator) (97.0%) were purchased from Aladdin Biochemical Technology Co., Ltd. (Shanghai, China). Graphite paper with a thickness of 200 μm was purchased from Jinglong Carbon

Technology Co., Ltd. (Beijing, China). Deionized water with a resistivity of 18.2 M Ω cm was used throughout the experiments.

The mixture was injected into a sandwich module with a certain size and irradiated with ultraviolet light for 3 h to produce the hydrogel. The power of the ultraviolet lamp is 50 W, and the height of the sample from the ultraviolet lamp is 20 cm.

The I₃⁻/I⁻ solution was prepared by dissolving KI and I₂ (molar ratio 2:1) in deionized water. The power of the ultrasonic cleaner is 20 W, and the ultrasound lasts for 8 h to completely dissolve KI and I₂. After the reaction of I⁻ and I₂ to form I₃⁻, the molar ratio of I⁻ and I₃⁻ was 1:1. The hydrogel i-TEC was prepared by immersing the hydrogel in the I₃⁻/I⁻ solution for 1 h. The residual solution was removed from the hydrogel surface by air flow.

5. *The experimental setup shown in Figure S6 was used for measuring Seebeck coefficients. Was this measurement performed in a vacuum or inert gas atmosphere? Were standard samples, such as constantan or nickel metal, used to verify the measurement reliability?*

Response: Thanks for your helpful comment. In our work, the measurement of Seebeck coefficients is carried out in a real environment (25°C, 75% HR). Thermoelectric materials are prone to chemical reactions with oxygen or other gases in the air at high temperatures. Oxidation can alter the properties of materials, thereby affecting test results. The use of vacuum or inert gas environment can avoid these reactions and ensure the accuracy and reliability of test results. The thermoelectric material we prepared operates at a low temperature difference and does not react significantly with gases in the air, so the testing process does not need to be carried out in a vacuum or inert atmosphere. There are many reports of collecting low-grade heat by ion thermoelectric materials in the air at room temperature (Science 368, 6495, (2020); Nat. Commun. 15, 6704, (2024); Nat. Commun. 14, 3246, (2023); Energy Environ. Sci. 15, 2974-2982 (2022)). To ensure the reliability of the measurement, we used 5mM I₃⁻/I⁻ aqueous solution as a standard sample to calibrate the test setup.

In response to this comment, we added two sentences in the caption of figure S6 to

clarify the experimental details.

The measurement is carried out in a real environment (25°C, 75% relative humidity).

To ensure the reliability of the measurement, we used 5mM I₃⁻/I⁻ aqueous solution as a standard sample to calibrate the test device.

6. *The output measurement of the device prepared by coupling n- and p-type materials is unclear. The author should include a schematic or actual image of the device, along with details of any standard instruments used. While the extraordinary output performance with such a small temperature gradient is impressive, a comparison with the output power of flexible inorganic/organic materials tested under similar conditions for low-grade waste heat recovery would strengthen the findings. The author could incorporate this comparison using a bar diagram or table, referencing relevant studies. (1) ACS Appl. Mater. Interfaces 2022, 14, 34, 38642–38650 (2) ACS Appl. Mater. Interfaces 2023, 15, 40, 46962–46970*

Response: We sincerely appreciate the reviewer for this constructive suggestion to strengthen the manuscript. In response to this suggestion, we added a schematic image of a homemade Seebeck coefficient measurement device (Figure S22). The heating plate was located below the tested material to provide a temperature difference. The thermal voltage was recorded using an electrochemical workstation (CHI760E). The temperature at the hot and cold sides was recorded using a thermocouple (Benetech GM1312).

Figure S22. Schematic image of a homemade Seebeck coefficient measurement device. The heating plate was located below the tested material to provide a temperature difference. The thermal voltage was recorded using an electrochemical workstation (CHI760E). The temperature at the hot and cold sides was recorded using a thermocouple (Benetech GM1312).

Ionic thermoelectric materials and electronic thermoelectric materials are two types of thermoelectric materials, with significant differences in their conduction mechanisms and properties. (1) Transmission mechanism: Ionic thermoelectric materials mainly involve the migration and diffusion of ions, which usually occur in solid or liquid media; Electronics thermoelectric materials mainly involve the transport of electrons, which usually occurs in metals, semiconductors or conductive polymers thermoelectricity. (2) Performance: Ionic thermoelectric materials typically have lower thermal conductivity, higher Seebeck coefficient and lower electrical conductivity, which means that they can generate higher thermoelectric voltages; Electronic thermoelectric materials typically have high electrical conductivity and low Seebeck coefficient and high thermal conductivity, which enable them to more efficiently convert thermal energy into electrical energy. Normally, comparisons between ion and electron thermoelectric materials are not made. Compared with other series integrated hydrogel i-TECs, the thermoelectric device we prepared has high output voltage and power (Table S2).

Table S2. Comparison of thermoelectric performance of series integrated hydrogel i-TECs

Electrolyte	VOC (V)	P_{max} (μ W)	Cell number (n)	Ref.
MAA/DMAPS/I ₃ ⁻ /I ⁻ /KCl	1.8	85	20	This work
I ₃ ⁻ /I ⁻ /PNIPAM nanogel	1	9	100	2
Gelatin-KCl-FeCN ^{3-/4-}	2.2	5	25	8

PVA/FeCN ^{3-/4-}	0.7	0.3	118	9
PVA/Fe ^{3+/2+} /CMC/FeCN ^{3-/4-}	0.34	39	36	10
PAAm/Fe ^{3+/2+} /PAAm/FeCN ^{3-/4-}	0.16	0.77	28	11
Gelatin-FeCN ^{3-/4-} /Gr	0.12	4.2	20	12
Gelatin-KCl-FeCN ^{3-/4-}	2.8	68	24	13

In response to this suggestion, we added Table S2 and a sentence to compare the thermoelectric performance of series integrated hydrogel i-TECs.

Reference:

- 1 Abraham, T. J., MacFarlane, D. R., Baughman, R. H., Jin, L. Y., Li, N. & Pringle, J. M. Towards ionic liquid-based thermoelectrochemical cells for the harvesting of thermal energy. *Electrochim. Acta* **113**, 87-93 (2013).
- 2 Duan, J. J., Yu, B. Y., Liu, K., Li, J., Yang, P. H., Xie, W. K., Xue, G. B., Liu, R., Wang, H. & Zhou, J. P-N conversion in thermogalvanic cells induced by thermo-sensitive nanogels for body heat harvesting. *Nano Energy* **57**, 473-479 (2019).
- 3 Zhou, H. Y., Yamada, T. & Kimizuka, N. Supramolecular thermo-electrochemical cells: enhanced thermoelectric performance by host-guest complexation and salt-induced crystallization. *J. Am. Chem. Soc.* **138**, 10502-10507 (2016).
- 4 Han, Y., Zhang, J., Hu, R. & Xu, D. Y. High-thermopower polarized electrolytes enabled by methylcellulose for low-grade heat harvesting. *Sci. Adv.* **8**, eabl5318 (2022).
- 5 Kim, K., Kang, J. & Lee, H. Hybrid thermoelectrochemical and concentration cells for harvesting low-grade waste heat. *Chem. Eng. J.* **426**, 131797 (2021).
- 6 Wang, H., Zhuang, X. Y., Xie, W. K., Jin, H. R., Liu, R., Yu, B. Y., Duan, J. J., Huang, L. & Zhou, J. Thermosensitive-CsI₃-crystal-driven high-power I⁻/I₃⁻ thermocells. *Cell Rep. Phys. Sci.* **3**, 100737 (2022).
- 7 Liang, Y. M., Hui, J. K. H., Morikawa, M. A., Inoue, H., Yamada, T. & Kimizuka, N. High positive Seebeck coefficient of aqueous I⁻/I₃⁻ thermocells based on host-guest interactions and LCST behavior of PEGylated α -cyclodextrin. *ACS Appl. Energ. Mater.* **4**, 5326-5331 (2021).
- 8 Han, C. G., Qian, X., Li, Q. K., Deng, B., Zhu, Y. B., Han, Z. J., Zhang, W. Q., Wang, W. C., Feng, S. P., Chen, G. & Liu, W. S. Giant thermopower of ionic gelatin near room temperature. *Science* **368**, 1091-1098 (2020).
- 9 Yang, P. H., Liu, K., Chen, Q., Mo, X. B., Zhou, Y. S., Li, S., Feng, G. & Zhou, J. Wearable thermocells based on gel electrolytes for the utilization of body heat. *Angew. Chem. Int. Ed.* **55**, 12050-12053 (2016).
- 10 Liu, Y. Q., Zhang, S., Zhou, Y. T., Buckingham, M. A., Aldous, L., Sherrell, P. C., Wallace, G. G., Ryder, G., Faisal, S., Officer, D. L., Beirne, S. & Chen, J. Advanced wearable thermocells for body heat harvesting. *Adv. Energy Mater.* **10**, 2002539 (2020).
- 11 Xu, C., Sun, Y., Zhang, J. J., Xu, W. & Tian, H. Adaptable and wearable thermocell based on stretchable hydrogel for body heat harvesting. *Adv. Energy Mater.* **12**, 2201542 (2022).
- 12 Han, C. G., Zhu, Y. B., Yang, L. J., Chen, J. W., Liu, S. J., Wang, H. Y., Ma, Y. M., Han, D. X. &

- Niu, L. Remarkable high-temperature ionic thermoelectric performance induced by graphene in gel thermocells. *Energy Environ. Sci.* **17**, 1559-1569 (2024).
- 13 Li, Y. C., Li, Q. K., Zhang, X. B., Deng, B., Han, C. G. & Liu, W. S. 3D hierarchical electrodes boosting ultrahigh power output for gelatin-KCl-FeCN^{4/3-} ionic thermoelectric cells. *Adv. Energy Mater.* **12**, 2103666 (2022).

Point-by-Point Response to the Reviewer 2's comments

Author have revised manuscript. I have no further questions.

Response: We are grateful to the reviewer for his/her positive and encouraging feedback. We also sincerely appreciate the reviewer for his/her constructive comments, which help improve the quality of our manuscript.